# Water-mediated deracemization of a bisporphyrin helicate assisted by diastereoselective encapsulation of chiral guests

Naoki Ousaka[1,4], Shinya Yamamoto[1], Hiroki Iida [1,5], Takuya Iwata[1], Shingo Ito[2], Yuh Hijikata [2,3], Stephan Irle[2,3,6] & Eiji Yashima [1,4]

Deracemization is a powerful method by which a racemic mixture can be transformed into an excess of one enantiomer with the aid of chiral auxiliaries, but has been applied only to small chiral molecular systems. Here we report a deracemization of a racemic double-stranded spiroborate helicate containing a bisporphyrin unit upon encapsulation of chiral aromatic guests between the bisporphyrin. The chiral guest-included helicate is kinetically stable, existing as a mixture of right- and left-handed double helices, which eventually undergo an inversion of the helicity triggered by water resulting from the water-mediated reversible diastereoselective B-O bond cleavage/reformation of the spiroborate groups, thus producing an optically-active helicate with a high enantioselectivity. Quantum chemical calculations suggest that the stereospecific CH-$\pi$ interactions between the porphyrin hydrogen atoms of the helicate and an aromatic pendant group of the chiral guest play a key role in the enhancement of the helical handedness of the helicate.

[1] Department of Molecular Design and Engineering, Graduate School of Engineering, Nagoya University, Nagoya 464-8603, Japan. [2] Department of Chemistry, Graduate School of Science, Nagoya University, Nagoya 464-8602, Japan. [3] Institute of Transformative Bio-Molecules (WPI-ITbM), Nagoya University, Nagoya 464-8601, Japan. [4] Present address: Department of Molecular and Macromolecular Chemistry, Graduate School of Engineering, Nagoya University, Nagoya 464-8603, Japan. [5] Present address: Department of Chemistry, Graduate School of Natural Science and Technology, Shimane University, 1060 Nishikawatsu, Matsue 690-8504, Japan. [6] Present address: Computational Sciences & Engineering Division, Oak Ridge National Laboratory, Oak Ridge 37831-6493 TN, USA. Correspondence and requests for materials should be addressed to E.Y. (email: yashima@chembio.nagoya-u.ac.jp)

The double-helix is one of the topologically unique structures, which is instantly reminiscent of the DNA double-helix. Hence, the design and synthesis of artificial double helices has become an attractive challenge with implications for sophisticated biological structures and functions[1–6]. Among the double helices prepared so far, helicates, a class of metal-directed self-assembled helical complexes, are the most popular structural motifs since the seminal work by Lehn and co-workers[7–11]. The control of its handedness is of key importance for developing one-handed helices with a specific functionality that involves separation[12] and sensing enantiomers[12–14] and asymmetric catalysis[15] as well as antimicrobial materials[16]. However, double-stranded helices composed of achiral molecular strands assembled with labile metal ions, such as Cu(I) ions, mostly exist in an equal mixture of interconvertible right- (P) and left-handed (M) helices and their resolution into the enantiomers remains difficult except for one example[17]. A preferred-handed helicity can be biased in such kinetically labile double-stranded helicates using a chiral template[18] or chiral auxiliaries[19–21], but the helicates likely lose their optical activities after removal of the chiral template or chiral auxiliaries due to irreversible racemization in solution. This asymmetric transformation of a racemic mixture into a nonracemic one by the presence of chiral species, a typical class of deracemizations[22,23], is known as the Pfeiffer effect[24], and was first observed in kinetically labile racemic coordination complexes an almost century ago[25,26]. Thereafter, this (asymmetric transformation strategy) concept has been applied to a variety of configurationally labile enantiomers[22,27–29] including helicates[19–21] and dynamically racemic helical polymers[30]. In contrast to kinetically labile helicates, double- and triple-stranded helicates with a hexa-coordinated, octahedral geometry composed of a substitution-inert metal are kinetically stable and can be readily separated into the enantiomeric helices by traditional resolution in the maximum yield of 50%[10,17,31,32]. Therefore, it is significantly required to develop a versatile method, which enables to quantitatively deracemize a racemic mixture of kinetically labile helicates into kinetically stable helicates with an excess of one enantiomer upon noncovalent interactions with chiral guests. This deracemization technique is an ideal and promising approach to produce an excess of one enantiomer or a nonracemic product in 100% theoretical yield from a racemic mixture[23,33–35], but has been limited to small chiral molecular systems[33–35] except for one example observed in supramolecular helical polymers[36], which are stereochemically stable but can deracemize via a bond cleavage/reformation process that proceeds in a diastereoselective or enantioselective fashion assisted by chiral catalysts in dynamic kinetic resolutions[37] and chiral hosts or upon crystallization with chiral external forces[35,38], thus producing nonracemic products or enantiomers.

In an earlier study, we reported a series of racemic double-stranded helicates consisting of two spiroborate groups bridged by two achiral tetraphenol strands bearing a variety of linkers in the middle[39]. The racemic helicates can be resolved into optically-pure enantiomers by conventional diastereomeric salt formation followed by ion exchange with an achiral ammonium salt[39–41]. The optically-active spiroborate helicates are stable and tolerant toward racemization in aprotic polar solvents, but racemize in the presence of a catalytic amount of acid as a proton source, indicating that the spiroborate helicates possess both dynamic (labile) and static (inert) features toward racemization (Fig. 1b).

Here, we show a water-mediated deracemization of a racemic spiroborate helicate containing a bisporphyrin[41–46] unit (1)[47] (Fig. 1a) upon diastereoselective encapsulation of an electron-deficient chiral aromatic guest between the porphyrin rings of the racemic helicate (Fig. 1c) assisted by stereospecific CH–π interactions. The interconversion between the (P)- and (M)-helicates

takes place through a water-mediated dynamic B–O bond[48,49] cleavage/reformation reaction that occurred at the spiroborate moieties in a highly helix-sense selective way, thus producing an optically-active *static* helicate from a racemic mixture after the removal of water.

## Results

### Synthesis of the helicate and guests.
The racemic and enantiopure (P)- or (M)-bisporphyrin helicates (rac-$1_{Na2}$ and (P)- or (M)-$1_{TBA2}$, respectively; TBA = tetra-n-butylammonium, Fig. 1a) were prepared according to a previously reported method[47]. The helicate 1 forms a stable inclusion complex with an electron-deficient planar guest, such as G1 ($1_{Na2}$⊃G1) (Fig. 1a), sandwiched between the porphyrins with a high association constant ($K_a$ = ca. $2.2 \times 10^9$ M$^{-1}$ in CH$_3$CN). Thus, a series of naphthalenemonoimide (NMI)-based enantiopure and/or racemic guests G2–G8 (Fig. 1a) were synthesized by the reaction of G1 with various kinds of chiral primary amines (see Supplementary Methods).

### Water-mediated racemization of the helicate.
The water-mediated racemization kinetics of (M)-$1_{TBA2}$ was first investigated by circular dichroism (CD) spectroscopy in aprotic polar solvents such as dimethyl sulfoxide (DMSO) (Supplementary Fig. 1). As anticipated, the intense positive-bisignate CD signal of (M)-$1_{TBA2}$[36] remained unchanged after heating at 70 °C for 24 h in anhydrous DMSO, whereas its CD intensity gradually decreased with time at high temperatures (70–110 °C) as a result of the racemization in the presence of water (ca. 500 equivalents to (M)-$1_{TBA2}$) (Supplementary Fig. 1a,b). The pseudo-first-order rate constants ($k_{rac}$, s$^{-1}$) and half-life time periods ($t_{1/2}$, h) for the racemization of (M)-$1_{TBA2}$ estimated by the CD intensity changes at different temperatures provided the thermodynamic parameters based on the Arrhenius and Eyring plots of the kinetic data (Supplementary Fig. 1c–e), which are summarized in Supplementary Table 1. Interestingly, the inclusion complex formation of (M)-$1_{TBA2}$ with G1 resulted in an increase in the racemization rates of (M)-$1_{TBA2}$⊃G1 as compared to those of the free (M)-$1_{TBA2}$ under the same conditions (Supplementary Fig. 2), probably due to the B–O bond strain at the spiroborate moieties of the helicate[47] that increases upon complexation with G1. Such steric strain within the spiroborate helicate could be relaxed during the water-catalyzed B–O bond cleavage reactions at the spiroborate groups, leading to a significant increase in the activation entropy ($\Delta S^{\ddagger}$) for the racemization from $-42 \pm 11$ ((M)-$1_{TBA2}$) to $4 \pm 20$ J mol$^{-1}$ ((M)-$1_{TBA2}$⊃G1) (Supplementary Table 1). Intermolecular hydrogen bond formation between the included G1 and water, which may accelerate the water-catalyzed racemization of the (M)-$1_{TBA2}$⊃G1 inclusion complex, could also be taken into consideration. We note that inversion of the helicity of the helicate 1 requires simultaneous cleavages of one of the four B–O bonds at each spiroborate group by the formation of an achiral *meso* intermediate, followed by reforming of the spiroborated (P)- or (M)-helicate (Supplementary Fig. 1f).

### Helix-sense-selective deracemization of racemic helicate.
The deracemization of rac-$1_{Na2}$ was then performed in the presence of various chiral guests (3 equivalents) ((R)- or (S)-G2–G8, Fig. 1a) in various solvents containing a small amount of water at 80 °C for an appropriate length of time until reaching an equilibrium state. We chose this temperature (80 °C) based on the relationship between the diastereomeric excess (d.e.) of the optically-active $1_{Na2}$ (derac-$1_{Na2}$) complexed with (S)-G2 in DMSO-$d_6$ and the time required to reach an equilibrium state at various temperatures (Supplementary Fig. 3), although higher d.e. values will be

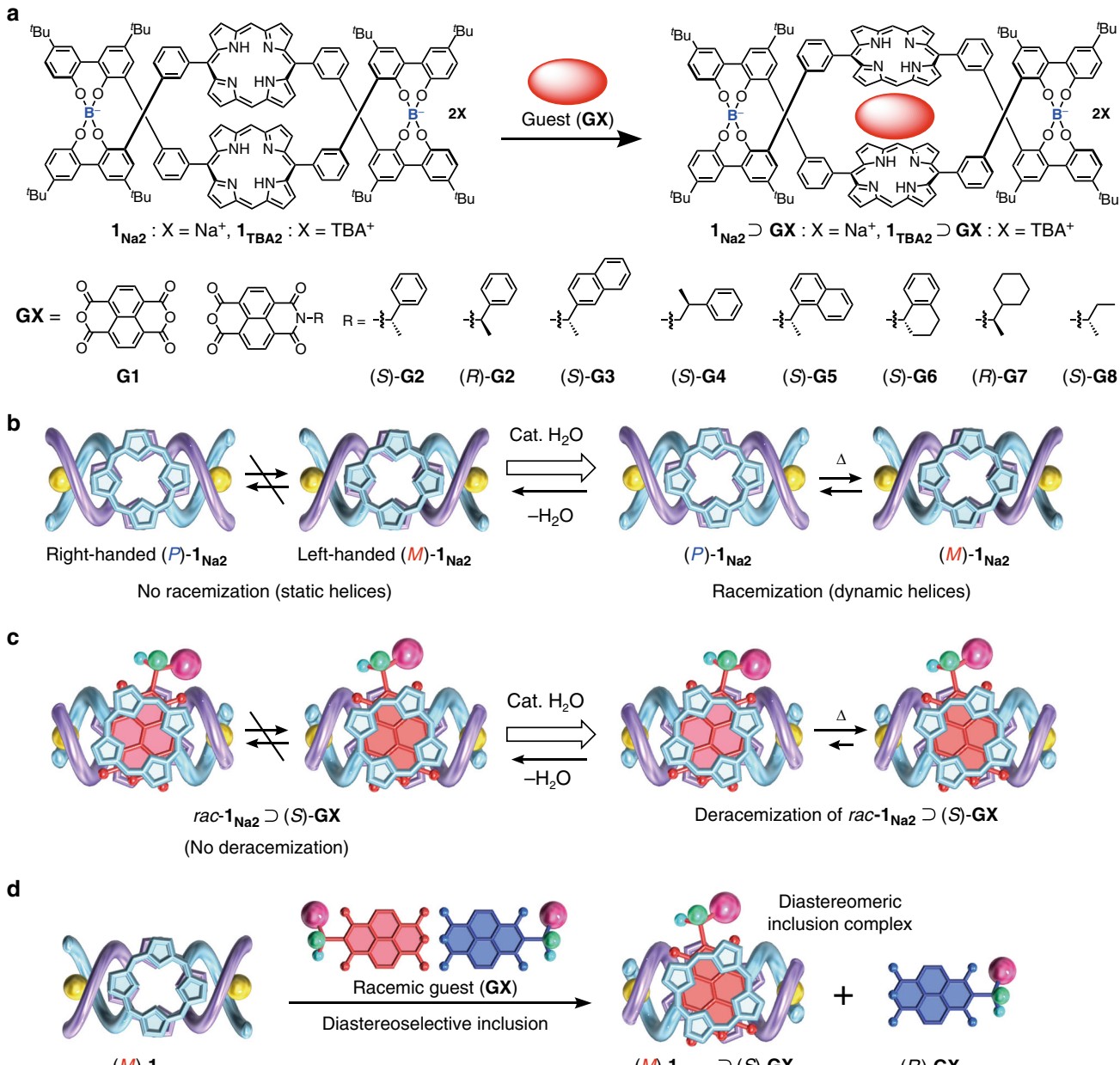

**Fig. 1** Deracemization and diastereoselective inclusion complexation of double-stranded spiroborate helicates. **a** Chemical structures of double-stranded bisporphyrin helicates **1$_{X2}$** (X = Na⁺ or TBA⁺) and their inclusion complexation with achiral (**G1**) or chiral (**G2**–**G8**) electron-deficient aromatic guests. **b**, **c** Schematic representations of water-catalyzed racemization of *rac*-**1$_{Na2}$** (**b**), deracemization of *rac*-**1$_{Na2}$** upon the inclusion complex formation with an enantipure guest (**c**), and diastereoselective inclusion complexation of racemic guest with left-handed (*M*)-**1$_{TBA2}$** (**d**)

obtained by a prolonged heating at lower temperatures. The d.e. values of *derac*-**1$_{Na2}$** complexed with chiral guests ((*P*)- and (*M*)-**1$_{Na2}$**⊃chiral guest) were determined based on their ¹H nuclear magnetic resonance (NMR) spectra except for those with **G6** and **G7** (Table 1, runs 1–8 and Fig. 2a and Supplementary Fig. 4). The chiral guests included in the *derac*-**1$_{Na2}$** were then replaced by the achiral **G1** to produce the corresponding enantiomeric *derac*-**1$_{Na2}$**⊃**G1** complexes (for detailed experimental procedures, see Methods) and their enantiomeric excess (e.e.) values that correspond to their helical sense excesses of the *derac*-**1$_{Na2}$** and its helical handedness (*P* or *M*) were determined by their CD spectra (Fig. 2b) based on the molar ellipticity at 419 nm ($\Delta\varepsilon_{419}$) and its sign of the one-handed helical (*M*)-**1$_{TBA2}$**⊃**G1** complex (e.e. >99%)[47] as the base value, respectively (Table 1, runs 1–8). The complete replacement of the guests with **G1** was confirmed by

model experiments (Supplementary Fig. 5). The CD spectra of the *derac*-**1$_{Na2}$**⊃**G1** complexes assisted by (*R*)- and (*S*)-**G2** displayed perfect mirror images with almost similar e.e. values of 44% and 47%, respectively, which are in good agreement with the corresponding d.e. values (44%) estimated by ¹H NMR (Table 1, runs 1 and 2).

The deracemization results of *rac*-**1$_{Na2}$** assisted by the chiral guests (**G2**–**G8**) in DMSO-*d₆* at 80 °C revealed that the helix-sense (*P* or *M*) and its helix-sense excess (% e.e.) of the produced *derac*-**1$_{Na2}$** were significantly affected by the structures of the chiral pendant groups of the **NMI**-based guests (Table 1, runs 1–8). The chiral guests bearing bulky aromatic pendants, such as the 1-phenylethtyl ((*S*)- and (*R*)-**G2**) and 1-(2-naphthyl)ethyl ((*S*)-**G3**) groups, afforded an optically-active *derac*-**1$_{Na2}$** with an appreciable level of helix-sense selectivity at 80 °C (47% and 55%

**Table 1 Deracemization of *rac*-1$_{Na2}$ with chiral guests in various solvents at 80 °C**

| Run | Guest | Solvent[a] | e.e.[b] [d.e.][c] (%) | Run | Guest | Solvent[a] | e.e.[b] [d.e.][c] (%) |
|---|---|---|---|---|---|---|---|
| 1 | (S)-**G2** | DMSO-$d_6$ | 47 [44] (M-rich) | 14 | (S)-**G4** | THF-$d_8$ | 29 [32] (M-rich) |
| 2 | (R)-**G2** | DMSO-$d_6$ | 44 [44] (P-rich) | 15 | (S)-**G2** | acetone-$d_6$ | 32 [34] (M-rich) |
| 3 | (S)-**G3** | DMSO-$d_6$ | 55 [53] (M-rich) | 16 | (S)-**G3** | acetone-$d_6$ | 31 [32] (M-rich) |
| 4 | (S)-**G4** | DMSO-$d_6$ | 10 [10] (M-rich) | 17 | (S)-**G2** | IBN | 44 (M-rich) |
| 5 | (S)-**G5** | DMSO-$d_6$ | 28 [27] (M-rich) | 18 | (S)-**G3** | IBN | 43 (M-rich) |
| 6 | (S)-**G6** | DMSO-$d_6$ | 6 [n.d.][d] (P-rich) | 19 | (S)-**G2** | DMAc | 33 (M-rich) |
| 7 | (R)-**G7** | DMSO-$d_6$ | 28 [n.d.][d] (M-rich) | 20 | (S)-**G3** | DMAc | 38 (M-rich) |
| 8 | (S)-**G8** | DMSO-$d_6$ | 2 [4] (P-rich) | 21 | (S)-**G2** | DMF | 47 (M-rich) |
| 9 | (S)-**G2** | CD$_3$CN | 16 [14] (M-rich) | 22 | (S)-**G3** | DMF | 50 (M-rich) |
| 10 | (S)-**G3** | CD$_3$CN | 12 [12] (M-rich) | 23 | (S)-**G2** | DEF | 59 (M-rich) |
| 11 | (S)-**G4** | CD$_3$CN | 9 [12] (P-rich) | 24 | (S)-**G3** | DEF | 64 (M-rich) |
| 12 | (S)-**G2** | THF-$d_8$ | 59 [58] (M-rich) | 25 | (S)-**G2** | DIPF | 69 (M-rich) |
| 13 | (S)-**G3** | THF-$d_8$ | 65 [62] (M-rich) | 26 | (S)-**G3** | DIPF | 72 (M-rich) |

[b]Estimated by CD after replacement of the included chiral guests with achiral **G1**
[c]Estimated by $^1$H NMR
[a]Dimethyl sulfoxide (DMSO), acetonitrile-$d_3$ (CD$_3$CN), tetrahydrofuran (THF), isobutyronitrile (IBN), N,N-dimethylacetamide (DMAc), N,N-dimethylformamide (DMF), N,N-diethylformamide (DEF), and N,N-diisopropylformamide (DIPF). These solvents contain a small amount of H$_2$O (5–150 equivalents)
[d]Could not be estimated by $^1$H NMR due to its complicated spectral pattern

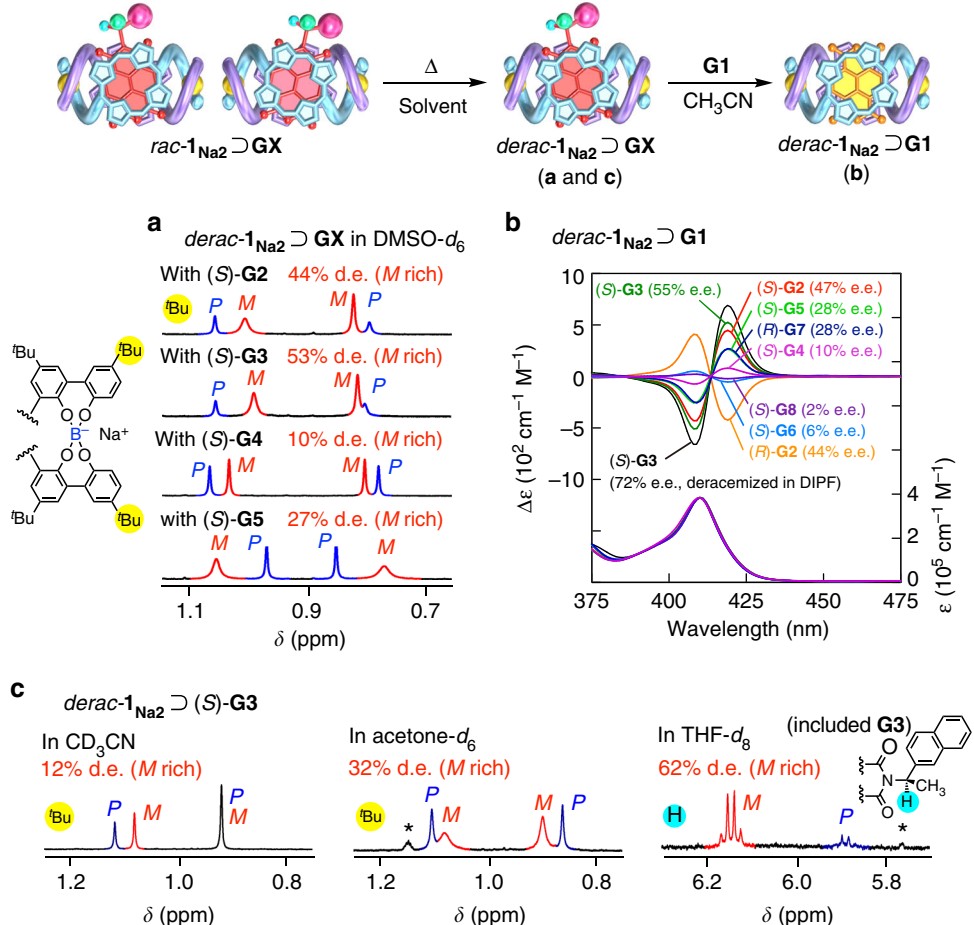

**Fig. 2** Deracemization of *rac*-1$_{Na2}$ upon inclusion complexation with various enantiopure guests in various solvents. **a** Partial $^1$H NMR spectra of mixtures of *rac*-1$_{Na2}$ (0.50 mM) with (S)-**G2**, (S)-**G3**, (S)-**G4**, and (S)-**G5** ([guest]/[1$_{Na2}$] = 3) in DMSO-$d_6$ measured at ambient temperature after heating at 80 °C for 24–48 h. [H$_2$O]/[1$_{Na2}$] = ca. 5–150. For the assignment of (P)- and (M)-helicity, see the text. **b** CD and absorption spectra of *derac*-1$_{Na2}$ (0.20 mM) in the presence of 3 equivalents of achiral **G1** measured in CH$_3$CN/DMSO-$d_6$ (ca. 29/1, v/v) or CH$_3$CN/DIPF (ca. 29/1, v/v) at 25 °C after deracemization of *rac*-1$_{Na2}$ upon inclusion complexation with **GX** ([guest]/[1$_{Na2}$] = 3) in DMSO-$d_6$ (for **G2**–**G8**) or DIPF (for (S)-**G3**) at 80 °C for 24–48 h. For detailed experimental procedures, see Methods (Procedure A). **c** Partial $^1$H NMR spectra of 1$_{Na2}$ (0.50 mM) with (S)-**G3** ([(S)-**G3**]/[1$_{Na2}$] = 3) in CD$_3$CN, acetone-$d_6$, and THF-$d_8$ measured at ambient temperature after heating at 80 °C for 26–266 h. [H$_2$O]/[(M)-1$_{TBA2}$] = ca. 5–20. Asterisk denotes the protons from unknown impurities

e.e., respectively), while those carrying aromatic, but flexible ((S)-G4) and aliphatic ((R)-G7 and (S)-G8) pendants produced the derac-$1_{Na2}$ with low % e.e. (10%, 28%, and 2%, respectively). Surprisingly, the chiral aromatic guests (S)-G5 and (S)-G6 that are similar to G3 and G2 in their structures afforded a slightly enantio-enriched derac-$1_{Na2}$ with 28% and 6% e.e., respectively. Among the chiral guests, (S)-G2–(S)-G5 produced the same (M)-rich derac-$1_{Na2}$, while (S)-G6, (R)-G7, and (S)-G8 afforded derac-$1_{Na2}$ with the opposite handedness. These results suggest that the NMI-based guests with moderately steric chiral aromatic pendant groups capable of interacting with the rac-$1_{Na2}$ in a diastereoselective fashion are required for achieving a highly helix-sense-selective deracemization of the rac-$1_{Na2}$ (see the next section).

We next investigated the solvent effect on the deracemization of rac-$1_{Na2}$ assisted by (S)-G2 and (S)-G3 using various solvents at 80 °C under identical conditions in DMSO-$d_6$ (Table 1). The helix-sense excesses of the produced derac-$1_{Na2}$ were estimated in the same way by CD and [1]H NMR (Fig. 2b, c and Supplementary Figs. 6 and 7).

Both of the guests preferentially afforded the (M)-rich derac-$1_{Na2}$ independent of the solvents (Table 1, runs 9, 10, 12, 13, 15–26), whereas the helix-sense-selectivities were highly dependent on the solvents, producing relatively high (M)-enriched derac-$1_{Na2}$ in THF, DMF, DEF, and DIPF (e.e. >50%) with up to 72% e.e. in DIPF when (S)-G3 was used as the guest (Table 1, run 26). Interestingly, the e.e. values of the derac-$1_{Na2}$ produced in similar solvents (DMF, DEF, and DIPF) increased with an increase in the N,N-disubstituted alkyl chain length (Table 1, runs 21–26). A similar increase in the e.e. value was also observed when IBN (43–44% e.e.) was used instead of CD$_3$CN (12–16% e.e.) as the solvent (Table 1, runs 9, 10, 17, and 18), indicating an important role of the solvent structures as well as its polarity.

The inclusion complex of rac-$1_{Na2}$⊃(S)-G3 could be completely separated into rac-$1_{Na2}$ and (S)-G3 by size-exclusion chromatography (SEC) using DMF as the eluent (Supplementary Fig. 8), which enables us to recover the optically-active $1_{Na2}$ for further applications, and at the same time, chiral guests for recycle. Encouraged by this straightforward separation, a scaled-up deracemization reaction of rac-$1_{Na2}$ (7.9 mg, 3.6 μmol) with (S)-G3 (3 equivalents) was performed in DIPF at 80 °C for 24 h, producing an optically-active derac-$1_{Na2}$⊃(S)-G3 with 74% d.e. (Supplementary Fig. 9). Subsequent SEC isolation afforded derac-$1_{Na2}$ and (S)-G3 in excellent yields of 91% and 93% with >70% and 99% e.e., respectively.

**Diastereoselective encapsulation of racemic guests**. We anticipated that the observed helix-sense-selective deracemization of the rac-$1_{Na2}$ with chiral guests was mostly due to differential inclusion complexation of an enantiopure guest ((S)-G2 for example) toward rac-$1_{Na2}$ with an appreciable level of diastereoselectivity. To confirm this and also to disclose an energetic driving force for the helix-sense-selectivity during the deracemization process, we investigated the chiral recognition ability[13,46,50–55] of the one-handed helical (M)-$1_{TBA2}$ toward racemic guests (G2–G4) in various solvents at different temperatures during the inclusion complexation in the bisporphyrin cavity of (M)-$1_{TBA2}$ by [1]H NMR spectroscopy (Fig. 1d).

Upon mixing (M)-$1_{TBA2}$ and (R)- or (S)-G2 in a 1:1 molar ratio in THF-$d_8$ at 25 °C, the 1:1 inclusion complex was quantitatively produced. The [1]H NMR signals of the (M)-$1_{TBA2}$ were split into two sets of new signals as a result of desymmetrization of the pseudo-$D_2$-symmetric structure of $1_{TBA2}$ upon the inclusion complexation with the non-symmetric NMI-based guests (Fig. 3a). Thus, a mixture of (M)-$1_{TBA2}$ with 3 equivalents of rac-G2 in THF-$d_8$ showed the signals

corresponding to the diastereomers (M)-$1_{TBA2}$⊃(R)-G2 and (M)-$1_{TBA2}$⊃(S)-G2 with 66% d.e. (S rich) (Fig. 3b), which is very consistent with the diastereoselectivity (73% d.e.) calculated by using the association constants of (M)-$1_{TBA2}$ with (S)-G2 (ca. $2.4 \times 10^6$ M$^{-1}$) and (R)-G2 (ca. $3.7 \times 10^5$ M$^{-1}$) estimated by the fluorescence titration experiments (Supplementary Fig. 10). The diastereoselectivity of (M)-$1_{TBA2}$ with rac-G2 was highly dependent on the solvents (Fig. 3b and Supplementary Fig. 12) being relevant to the deracemization results (Table 1), and the % d.e. value (S rich) at 25 °C drastically decreased in the following order: THF-$d_8$ (66) > DMF-$d_7$ (48) > acetone-$d_6$ (45) > DMSO-$d_6$ (39) > CD$_3$CN (8), accompanied by chirality inversion of the G2 preferentially included in (M)-$1_{TBA2}$ in CD$_3$CN (Fig. 3b and Supplementary Table 2), which is also consistent with the diastereoselectivity (4% d.e.) calculated by the association constants of (M)-$1_{TBA2}$ with (S)-G2 (ca. $5.9 \times 10^6$ M$^{-1}$) and (R)-G2 (ca. $6.4 \times 10^6$ M$^{-1}$) (Supplementary Fig. 11). Such a solvent-induced switching of the diastereoselectivity during the inclusion complexation of the racemic guests with (M)-$1_{TBA2}$ was also observed by changing the temperature (see below).

The diastereoselectivities of (M)-$1_{TBA2}$ with rac-G2 in THF-$d_8$ and CD$_3$CN were significantly dependent on the temperature (Supplementary Table 2), and the d.e. value gradually increased with the decreasing temperature and reached 94 (S-rich) and 42% (R-rich) at −75 and −35 °C, respectively, while the chirality of the enriched enantiomer of G2 in the (M)-$1_{TBA2}$ was inverted in CD$_3$CN at temperatures below 40 °C (Supplementary Fig. 13).

A similar temperature-dependent enhancement of the diastereoselectivity of (M)-$1_{TBA2}$ in THF-$d_8$ and CD$_3$CN together with inversion of the diastereoselectivity in CD$_3$CN was also observed for rac-G3 (Supplementary Fig. 14). Of particular interest is that (M)-$1_{TBA2}$ completely recognized the chirality of G3 enantiomers to form the inclusion complex only with (S)-G3 (>99% d.e.) in THF-$d_8$ at below −50 °C (Fig. 3c and Supplementary Table 2). In contrast, (M)-$1_{TBA2}$ was almost temperature-independent with moderate (R)- and (S)-selectivities to rac-G4 carrying an aromatic, but a flexible 2-phenylpropyl pendant group in CD$_3$CN and THF-$d_8$, respectively (Supplementary Fig. 15 and Supplementary Table 2). Thus, the solvent and chiral guest-dependent changes in the diastereoselectivity of (M)-$1_{TBA2}$ toward racemic chiral guests at high temperatures seem to be relevant to the deracemization results of rac-$1_{Na2}$ upon complexation with the enantiopure guests (Table 1, runs 9–14 and Supplementary Table 2).

The observed differences in the diastereoselectivities of (M)-$1_{TBA2}$ toward racemic guests G2–G4 indicated the important role of the position of the aromatic pendant groups in its diastereoselective inclusion complexation. We then constructed the inclusion model structures for a pair of the diastereomers, (M)-1 complexed with (R)- and (S)-G2 (Fig. 4a, b and Supplementary Figs. 17 and 18) as well as those with (R)- and (S)-G4 (Supplementary Fig. 19) based on the analogous crystal structure of the inclusion complex of rac-$1_{Na2}$⊃G1, in which the achiral symmetric G1 is sandwiched between the bisporphyrin in a parallel fashion via face-to-face stacking interactions[36], followed by the density functional theory (DFT) calculations (Supplementary Methods). The sandwich structures of (M)-$1_{TBA2}$⊃G2 and (M)-$1_{TBA2}$⊃G4 were supported by the [1]H two-dimensional (2D) NMR experiments (Supplementary Figs. 30–47). The locations of (R)- and (S)-G2 complexed with (M)-1 are best illustrated in Fig. 4a, b, in which the NMI cores are intercalated between the porphyrins with different orientations of the NMI cores. As a result, the pendant phenyl group of (S)-G2 is oriented almost perpendicular to the porphyrin rings (edge-to-face arrangement) so as to position above the meso-proton (meso-H (H$^{a'A}$)) and its neighboring β-proton (β-H: H$^{b2'A}$) of the porphyrin in Fig. 4a,

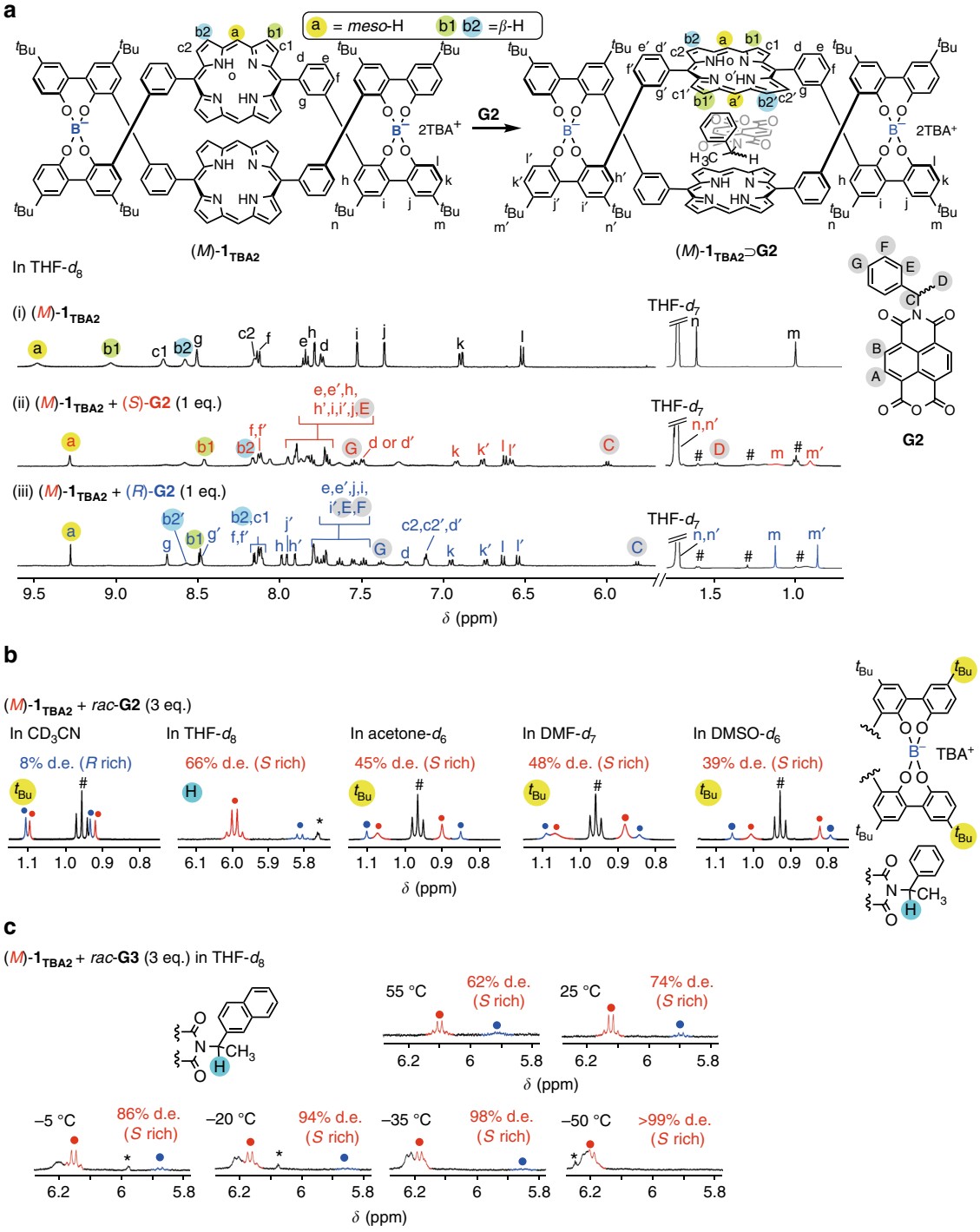

**Fig. 3** Diastereoselective inclusion complexation between (*M*)-**1**$_\text{TBA2}$ and *rac*-**G2** or *rac*-**G3**. **a** Schematic representations of diastereoselective inclusion complexation of racemic guest with left-handed (*M*)-**1**$_\text{TBA2}$. Partial $^1$H NMR spectra (500 MHz, 0.40 mM, 25 °C) of (*M*)-**1**$_\text{TBA2}$ in the absence (i) and presence of 1 equivalent of (*S*)-**G2** (ii), and (*R*)-**G2** (iii) in THF-$d_8$. **b** Partial $^1$H NMR spectra (500 MHz, 0.40 mM, 25 °C) of (*M*)-**1**$_\text{TBA2}$ in the presence of 3 equivalents of *rac*-**G2** in CD$_3$CN, THF-$d_8$, acetone-$d_6$, DMF-$d_7$, and DMSO-$d_6$. The diastereomeric excess (d.e.) values were estimated by the integral ratio of the $^t$Bu or methine proton signals derived from the diastereomeric inclusion complexes (*M*)-**1**$_\text{TBA2}$⊃(*S*)-**G2** (red closed circle) and (*M*)-**1**$_\text{TBA2}$⊃(*R*)-**G2** (blue closed circle). Hash and asterisk denote the protons from TBA and unknown impurities, respectively. The signals of (*M*)-**1**$_\text{TBA2}$, (*M*)-**1**$_\text{TBA2}$⊃(*R*)-**G2**, and (*M*)-**1**$_\text{TBA2}$⊃(*S*)-**G2** in THF-$d_8$ were assigned by two-dimensional gradient correlation spectroscopy (gCOSY) and rotating frame nuclear Overhauser enhancement spectroscopy (ROESY) measurements (Supplementary Figs. 20–35). **c** Variable temperature (VT) partial $^1$H NMR spectra (500 MHz, 0.40 mM) of (*M*)-**1**$_\text{TBA2}$ in the presence of 3 equivalents of *rac*-**G3** in THF-$d_8$ from −55 to 55 °C

although that of (*R*)-**G2** is positioned close to and above the *β*-protons (*β*-H: H$^{b1'A}$ and H$^{c1'A}$) instead of the H$^{a'A}$ and H$^{b2'A}$ protons (Fig. 4b). These spatial arrangements of the pendant phenyl groups suggest possible edge-to-face CH−π

interactions[56,57] between the pendant phenyl groups of (*S*)- and (*R*)-**G2** and the *meso*-H and/or *β*-H protons that can function more effectively for the (*M*)-**1**⊃(*S*)-**G2** complex, leading to the observed (*S*)-selective inclusion complexation with (*M*)-**1** in

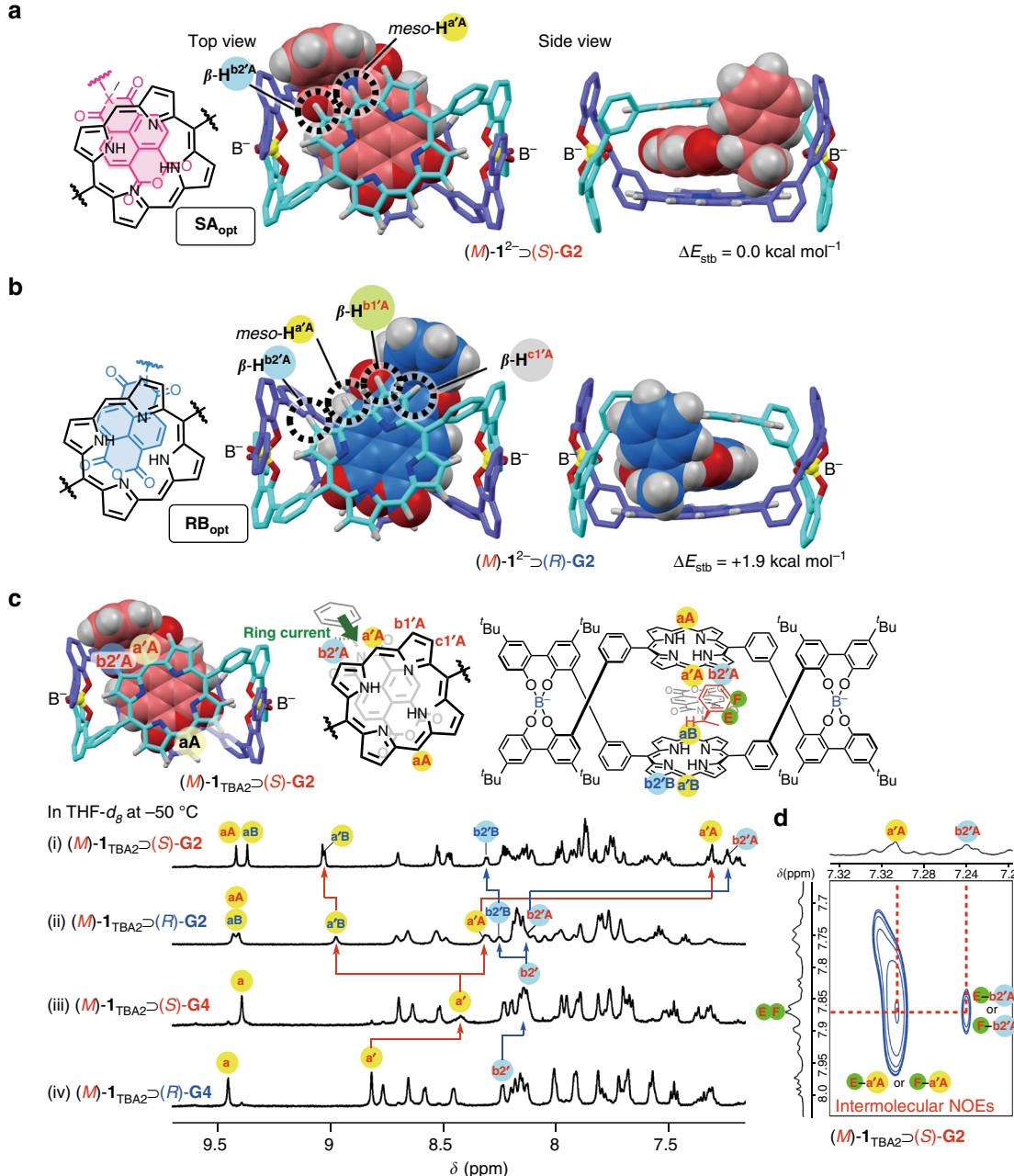

**Fig. 4** Mechanism of diastereoselective inclusion complexation of racemic guests with $(M)$-$\mathbf{1_{TBA2}}$. Top and side views of the energy-minimized inclusion complex structures of $(M)$-$\mathbf{1}^{2-}\supset(S)$-$\mathbf{G2}$ (**a**) and $(M)$-$\mathbf{1}^{2-}\supset(R)$-$\mathbf{G2}$ (**b**) with different geometries ($\mathbf{A_{opt}}$ and $\mathbf{B_{opt}}$) (see Supplementary Figs. 16 and 17), obtained by the DFT calculations with the D3 dispersion correction (for more details, see Supplementary Methods). The energy difference ($\Delta E_{stb}$) values are also shown (see also Supplementary Table 3). Hydrogen atoms except for *meso*- and $\beta$-protons of the porphyrin rings and $\mathbf{G2}$ are omitted for clarity. The $^t$Bu groups were replaced with hydrogen to simplify the calculations. The included guests are highlighted as a space-filling model. Hydrogen atoms except for *meso*- and $\beta$-protons of the porphyrin rings and $\mathbf{G2}$ are omitted for clarity. **c** Partial $^1$H NMR spectra (500 MHz, 0.40 mM, −50 °C) of $(M)$-$\mathbf{1_{TBA2}}$ in the presence of 1 equivalent of $(S)$-$\mathbf{G2}$ (i), $(R)$-$\mathbf{G2}$ (ii), $(S)$-$\mathbf{G4}$ (iii), and $(R)$-$\mathbf{G4}$ (iv) in THF-$d_8$. **d** Partial NOESY spectra (500 MHz, 0.40 mM, mixing time = 800 ms) of $(M)$-$\mathbf{1}^{2-}\supset(S)$-$\mathbf{G2}$ in THF-$d_8$ at −50 °C

various solvents except for CD$_3$CN at or below 40 °C (Fig. 3b and Supplementary Table 2). These energy-minimized models explain the significant upfield shifts of the specific *meso*-H and $\beta$-H protons of the porphyrin rings of $(M)$-$\mathbf{1_{TBA2}}$ complexed with $(S)$- and $(R)$-$\mathbf{G2}$ (Fig. 4c(i, ii) and Supplementary Figs. 24 and 25; see below) and are supported by the intermolecular nuclear Overhauser effect (NOE) cross-peaks observed between the *meso*-H and $\beta$-H protons of the porphyrin rings of $(M)$-$\mathbf{1_{TBA2}}$ and the aromatic protons of $(S)$-$\mathbf{G2}$ at low temperature (Fig. 4d and Supplementary Fig. 26).

In contrast, the phenyl groups of $(R)$- and $(S)$-$\mathbf{G4}$ are located far from the *meso*-H and $\beta$-H protons, so that effective CH–$\pi$ interactions may not be anticipated (Supplementary Fig. 19) as supported by slight upfield shifts of the *meso*-H and $\beta$-H protons compared to those of $(M)$-$\mathbf{1_{TBA2}}$ complexed with $\mathbf{G2}$ due to lack of the ring current effect of the pendant phenyl groups of $\mathbf{G4}$, thus showing moderate diastereoselectivities (18–38% d.e.) with opposite $(R)$- and $(S)$-$\mathbf{G4}$ selectivities in CD$_3$CN and THF-$d_8$, respectively (Supplementary Table 2). These results are in good agreement with those of the deracemization of *rac*-$\mathbf{1_{Na2}}$

complexed with (S)-**G4** in CD$_3$CN and THF-$d_8$, giving *derac*-**1**$_{Na2}$ with opposite *P* and *M* helicities, respectively (Table 1, runs 11 and 14).

This speculation was supported by the remarkable differences in the temperature-dependent $^1$H NMR spectral changes of (M)-**1**$_{TBA2}$ complexed between **G2** and **G4** in THF-$d_8$ (Supplementary Figs. 20–22). Upon complexation with the non-symmetric chiral **G2** and **G4**, all of the aromatic and porphyrin proton signals of (M)-**1**$_{TBA2}$ including the *meso*-H (a) and β-H (b$^1$ and b$^2$) signals of the porphyrin rings split into two sets of non-equivalent signals independent of the guests and solvents (CD$_3$CN and THF-$d_8$) at 25 °C (Fig. 3a and Supplementary Figs. 23 and 28) except for the (M)-**1**$_{TBA2}$⊃**G2** complex in THF-$d_8$; the specific *meso*-H (a') and β-H (b$^{1'}$ and b$^{2'}$) signals of the porphyrin rings were not observed at 25 °C due to broadening of the signals (Fig. 3a). In addition, the aromatic and porphyrin proton signals of the (M)-**1**$_{TBA2}$⊃(S)-**G2** complex in THF-$d_8$ were significantly broadened when compared to those of its diastereomer (M)-**1**$_{TBA2}$⊃(R)-**G2** (Fig. 3a) due to the relatively slow rotation of the (S)-1-phenylethyl group around the N–C$^α$ bond of the included (S)-**G2** at 25 °C (see below).

Upon cooling to a low temperature, the N–C$^α$ bond rotation of the (M)-**1**$_{TBA2}$⊃(S)-**G2** complex appeared to be further restricted from free rotation and finally frozen because more effective edge-to-face CH–π contacts act as a molecular brake[58] that prevents rotation around the N–C$^α$ bond of (S)-**G2** sandwiched between the two porphyrins, thereby leading to further splitting of the porphyrin and aromatic protons as well as the *tert*-butyl ($^t$Bu) protons of the helicate in THF-$d_8$ below −5 °C (Supplementary Fig. 21a; for proton peak splitting diagrams for the *meso*-H and β-H protons of (M)-**1**$_{TBA2}$ upon complexation with non-symmetric chiral guests, see Supplementary Fig. 20). Similar splitting of the proton resonances was also observed for the (M)-**1**$_{TBA2}$⊃(R)-**G2** complex, but appeared below −50 °C (Supplementary Fig. 21b). The chemical shift difference between the two newly-appeared *meso*-H signals (H$^{a'A}$ and H$^{a'B}$) (Δδ$_{meso}$) of the porphyrin rings of (M)-**1**$_{TBA2}$ complexed with (S)-**G2** (1.64 ppm at −25 °C) in THF-$d_8$ was significantly greater than that with (R)-**G2** (0.66 ppm at −50 °C), whereas the corresponding chemical shift difference between the β-H signals (H$^{b1'A}$ and H$^{b1'B}$) (Δδ$_β$) was ca. 0.04 ppm for (S)-**G2** at −25 °C that was smaller than 1.21 ppm for (R)-**G2** at −50 °C (Supplementary Figs. 24 and 25). These results are consistent with the calculated structural models (Fig. 4a, b) with respect to the relative orientations of the pendant phenyl groups of (R)- and (S)-**G2** complexed with (M)-**1**$^{2−}$. The 2D EXSY (exchange spectroscopy) spectra of the (M)-**1**$_{TBA2}$⊃(S)-**G2** complex acquired at different mixing times in THF-$d_8$ at −25 °C showed a series of chemical exchange cross-peaks between the two porphyrin protons due to slow rotation of the (S)-1-phenylethyl group (Supplementary Fig. 29), and the apparent exchange rate constant (rotation rate) was estimated to be 5.97 s$^{−1}$ (Supplementary Methods and Supplementary Fig. 29). In contrast, the (M)-**1**$_{TBA2}$⊃(R)-**G2** complex exhibited no chemical exchange cross-peaks in THF-$d_8$ at −25 °C due to fast rotation of the (R)-1-phenylethyl group, but showing the cross-peaks at −50 °C (Supplementary Fig. 25).

In sharp contrast, (R)- and (S)-**G4** bearing a methylene-linked flexible pendant did not show such non-equivalent splitting of the proton signals when complexed with (M)-**1**$_{TBA2}$ in THF-$d_8$ even at low temperatures because of almost the free rotation of the flexible pendant group (Supplementary Fig. 22), which clearly revealed the more favorable edge-to-face CH–π contacts of (M)-**1**$_{TBA2}$ with **G2** than those with **G4** in THF-$d_8$, thereby showing a much higher diastereoselectivity of (M)-**1**$_{TBA2}$ toward *rac*-**G2** than *rac*-**G4** in THF-$d_8$ (Supplementary Table 2). In addition, the N–C$^α$ bond rotation of the (M)-**1**$_{TBA2}$⊃(S)-**G2** complex became

much faster in CD$_3$CN comparable to that of the (M)-**1**$_{TBA2}$⊃(R)- and (S)-**G4** complexes in THF-$d_8$ judging from their variable-temperature $^1$H NMR spectral changes (Supplementary Figs. 22 and 27). Therefore, the diastereoselectivity of (M)-**1**$_{TBA2}$ toward *rac*-**G2** in CD$_3$CN significantly decreased, suggesting the important role of the CH–π interactions in the diastereoselective inclusion complexation[59] that are highly dependent on the solvents.

These $^1$H NMR analyses combined with the DFT calculated structures of the complexes revealed that the remarkable solvent- and guest-dependent changes in the diastereoselective inclusion complexation of the racemic guests in the (M)-**1**$_{TBA2}$ cavity appear to mainly rely on the edge-to-face CH-π interactions between the pendant phenyl rings of the guests and the protons at the *meso* and its neighboring β-positions of the porphyrin rings, because the CH–π interactions are sensitive to the solvents[60,61] and significantly contribute to the molecular and chiral recognition events[56,57,59].

According to the literature[60], the difference in the proton chemical shifts (Δδ$_{sol}$) of given solvents between in deuterated aromatic and non-aromatic solvents, such as C$_6$D$_6$ and CDCl$_3$, respectively, can be used as a measure to evaluate the CH–π interaction capabilities of the solvent molecules (Supplementary Table 5); the solvents with a higher Δδ$_{sol}$ value, such as CH$_3$CN, tend to strongly interact with aromatic molecules, and hence, the CH–π interactions between the host–guest complexes will be hindered. The difference in free energy (ΔΔG$_{inc}$) upon diastereomeric inclusion complexation between (M)-**1**$_{TBA2}$ and (S)-/(R)-**G2** in various solvents at 25 °C was then calculated using the d.e. value in each solvent (Supplementary Table 2) then plotted versus the Δδ$_{sol}$ values, which gave an almost straight line (Fig. 5a) (i), suggesting the very important role of the CH–π interactions in the diastereoselective inclusion complexation. For comparison, the ΔΔG$_{inc}$ values were also plotted versus other solvent parameters, such as the dipole moment (ii) and dielectric constant (iii), which did not show a good correlation (Fig. 5a).

In the same way, the calculated difference in free energy (ΔΔG$_{derac}$) in the helix-sense selective deracemization of *rac*-**1**$_{Na2}$ assisted by (S)-**G2** and (S)-**G3** in various solvents at 80 °C based on the d.e. values in Table 1 was plotted versus the Δδ$_{sol}$ values (i), dipole moment (ii), and dielectric constant (iii). Among them, the plot versus the Δδ$_{sol}$ values gave an approximate correlation between them (Fig. 5b), indicating the key role of the CH–π interactions being relevant to the diastereoselective inclusion complexation of enantiopure guests toward *rac*-**1**$_{Na}$. These results imply that if the deracemization experiment is carried out at lower temperatures in specific solvents, such as THF, the helix-sense selectivity may be significantly improved, although it will take a longer time to reach an equilibrium, while it may be possible when an acid instead of water is used as a catalyst for the catalytic B–O bond cleavage/reformation of the spiroborate groups of the helicate.

## Discussion

We have found that a racemic double-stranded spiroborate helicate deracemizes via water-mediated B–O bond cleavage/reformation of the spiroborate groups that proceeds in a highly helix-sense selective manner upon diastereoselective inclusion complexation of an enantiopure aromatic guest between the porphyrin rings linked to the racemic helicate. The double-stranded helicate is kinetically inert toward racemization in the absence of water, but its interconversion between the enantiomeric double helices can be switched on and off by the addition and removal of water. Attractive CH–π interactions between the

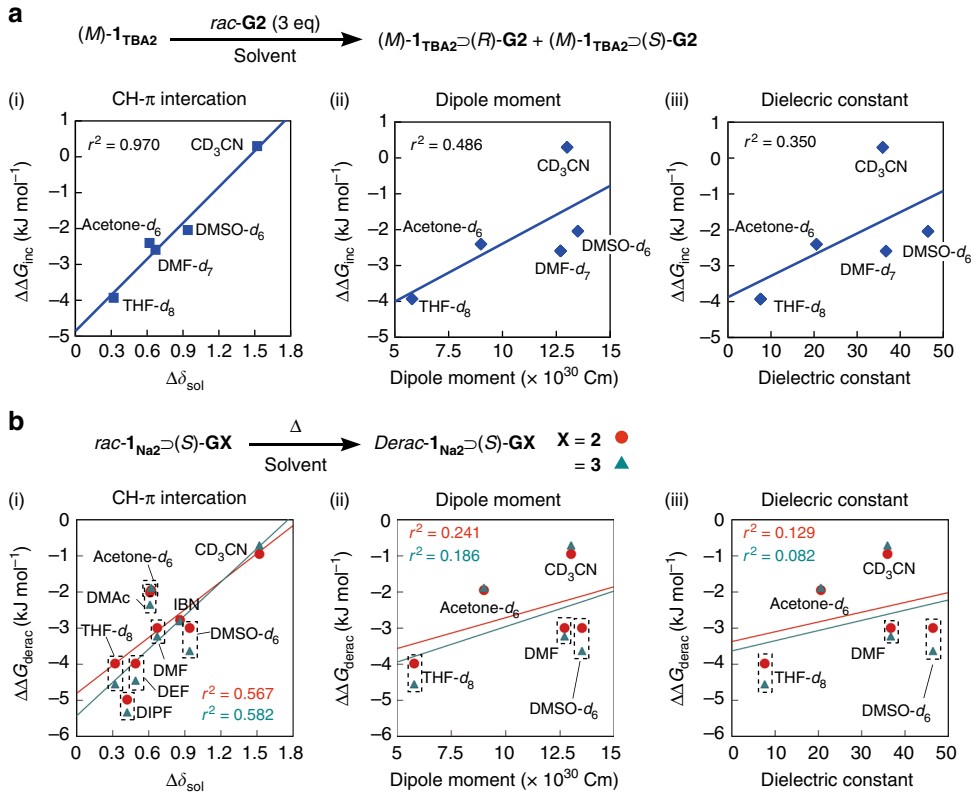

**Fig. 5** Mechanism of deracemization of *rac*-$1_{Na2}$ with chiral guests. **a** Plots of $\Delta\Delta G_{inc}$ (the difference in free energy for the diastereomeric inclusion complexation between (*M*)-$1_{TBA2}$ and (*S*)-**G2**/(*R*)-**G2** at 25 °C) against $\Delta\delta_{sol}$ (Supplementary Table 5) (i), dipole moment (ii), and dielectric constant (iii) of the solvents. **b** Plots of $\Delta\Delta G_{derac}$ (the difference in free energy for the diastereoselective deracemization of *rac*-$1_{Na2}$ with (*S*)-**G2** and (*S*)-**G3** at 80 °C) against $\Delta\delta_{sol}$ (i), dipole moment (ii), and dielectric constant (iii) of the solvents. Source data of (**a**) and (**b**) are provided as a Source Data file

pendant aromatic group of the encapsulated chiral guests and the porphyrin protons of the helicate are indispensable for an efficient diastereoselective inclusion complexation, thus leading to an excellent helix-sense selectivity during the water-mediated deracemization. The present findings imply that analogous dynamically racemic helicates and supramolecules can be converted into the corresponding kinetically-stable enantiomers via deracemization in the presence of chiral guests through noncovalent chiral interactions.

## Methods

**General procedures for the deracemization.** *Procedure A*: For the deracemization of *rac*-$1_{Na2}$ upon inclusion complexation with (*S*)-**G2** in DIPF, stock solutions of *rac*-$1_{Na2}$ (0.50 mM) (solution I) and (*S*)-**G2** (solution II) (2.0 mM) were prepared in CH$_3$CN. Aliquots of I (0.35 μmol, 700 μL) and II (1.05 μmol, 525 μL) were added to a vial, then the solvent was removed under reduced pressure. The vial containing *rac*-$1_{Na2}$ and 3 equivalents of (*S*)-**G2** was sealed with a rubber septum, and to this was added distilled DIPF (700 μL) using a syringe under nitrogen. The DIPF solution was then heated to 80 °C for the appropriate length of time (24 h) until reaching an equilibrium state. The H$_2$O content in the reaction mixture was estimated by measuring the $^1$H NMR spectrum of the mixture after dilution with dried DMSO-$d_6$ (DIPF/DMSO-$d_6$ = 1/70, v/v). The reaction progress was monitored at an appropriate time interval by CD and absorption measurements of the reaction mixture after cooling to room temperature, then diluted 30-fold with a CH$_3$CN solution containing 3 equivalents of the achiral **G1**. The percent e.e. value of the deracemized $1_{Na2}$ complexed with **G1** was then estimated based on the following equation:

$$e.e.(\%) = \Delta\varepsilon_{419}/\Delta\varepsilon_{419(max)} \times 100$$

where $\Delta\varepsilon_{419}$ and $\Delta\varepsilon_{419(max)}$ are the CD intensities (1st Cotton effect at 419 nm) of the *derac*-$1_{Na2}$⊃**G1** and enantiopure (*M*)-$1_{TBA2}$⊃**G1** (e.e. >99%), respectively.

In the same way, the deracemization of *rac*-$1_{Na2}$ upon inclusion complexation with enantiopure guests in various solvents were performed and the results are summarized in Table 1.

*Procedure B*: The deracemization progress was also directly monitored by $^1$H NMR spectroscopy in CD$_3$CN, THF-$d_8$, DMSO-$d_6$, and acetone-$d_6$. For the deracemization of *rac*-$1_{Na2}$ upon inclusion complexation with **G2** in DMSO-$d_6$, stock solutions of *rac*-$1_{Na2}$ (0.50 mM) (solution I) and (*S*)-**G2** (solution II) (2.0 mM) were prepared in CH$_3$CN. Aliquots of I (0.35 μmol, 700 μL) and II (1.05 μmol, 525 μL) were added to an NMR tube, then the solvent was removed under reduced pressure. The NMR tube was sealed with a rubber septum, then subjected to three evacuation/nitrogen fill cycles. To this was added 700 μL of DMSO-$d_6$ via a syringe. The tube was then sealed with a small flame and the solution was heated to 80 °C for an appropriate length of time (38 h) until reaching an equilibrium state. The reaction progress was monitored at an appropriate time interval by $^1$H NMR measurements of the reaction mixture after cooling to 25 °C. The d.e. value of the deracemized $1_{Na2}$ was estimated by the integral ratio of the diastereomeric $^t$Bu signals derived from (*M*)-$1_{Na2}$⊃(*S*)-**G2** and (*P*)-$1_{Na2}$⊃(*S*)-**G2** (Fig. 2a). In the same way, the deracemization of *rac*-$1_{Na2}$ upon inclusion complexation with enantiopure guests in various solvents, such as CD$_3$CN, acetone-$d_6$, and THF-$d_8$, was performed at 80 °C and the results are summarized in Table 1.

## Data availability

The authors declare that the data supporting the findings of this study are available within the paper and its Supplementary Information files, and from the corresponding authors upon reasonable request. Source data underlying Fig. 5a, b and Supplementary Figs. 1b–e, 2a–d, 3, 10b,d, 11b,d, and 29b are provided as a Source Data file.

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

## Acknowledgements

The authors thank Daisuke Taura, Akio Urushima, and Manabu Itakura (Nagoya University) for their help in the experimental assistance and also Hiroyuki Asanuma and Hiromu Kashida (Nagoya University) for providing advice regarding the fluorescence quenching. This work was supported in part by JSPS KAKENHI (Grant-in-Aid for Scientific Research (S), No. 25220804 (E.Y.), Grant-in-Aid for Specially Promoted Research, No. 18H05209 (E.Y.), and Grant-in-Aid for Young Scientists (B), No. 17K14470 (N.O.)). S.Y. expresses his thanks for a JSPS Research Fellowship for Young Scientists (No. 9119).

## Author contributions

E.Y. conceived and directed the project. N.O., H.I., and E.Y. designed the experiments. S.Y. and T.I. performed all of the experiments. S. Ito, Y.H., and S. Irle performed the calculations. N.O. and E.Y. analyzed the data and co-wrote the paper. All authors discussed the results and edited the manuscript.

## Additional information

**Competing interests:** The authors declare no competing interests.

