## [Peer Review File · Nature Communications]

Reviewers' comments:

Reviewer #1 (Remarks to the Author):

I find it hard to evaluate the novelty of this work. The manuscript claims the process of deracemization and specifically the use of water as the activator of this process to be the original element of this work. They make further reference to the double helix as a shape related to DNA and therefore glean the luster coming from all of DNA's biological implications.

What I like is the complexity of the system and that it seems to work efficiently. There is something appealing about these kinds of structures.

That said, what I don't like is the justification. Deracemization as a process is as old as basic 1st and 2nd order asymmetric transformations. That these occur selectively in the presence of chiral agents was postulated at least as early as Quehl and Pfeiffer's work on chiral coordination complexes in *Berichte*, 1931, 64, 2667 and by Kuhn for biaryls in *Berichte* 1932 65, 49. These cases were of labile systems shifted to one form, but the degree of lability could be easily imagined as tunable and perhaps has been in the intervening 85 years.

Norden showed in the 1986 the deracemization of iron complexes on DNA, see: Härd T, Nordén B. Enantioselective interactions of inversion-labile trigonal iron(II) complexes upon binding to DNA. *Biopolymers*. 1986 Jul;25(7):1209-28. Thus the idea of using DNA or double-helix forms has been broached.

So I don't see the authors as having made a milestone leap in the concept of deracemization, but perhaps the peculiarity of borate esters as the stability element and water and the tunable factor make this of special interest to readers.

At a minimum, I would hope the authors would teach the reader more about 1st order asymmetric transformations and their history dating back a century. I would ask that they downplay statements claiming that deracemization is somehow new or only applied to very few systems. Indeed, they should highlight that the energy difference only has to be a couple of kcal/mol so in polymers this might be even easier to achieve.

Once a proper context is laid out, then their novel water switching mechanism will probably tickle some reader's fancy.

Reviewer #2 (Remarks to the Author):

The manuscript by Yashima et al. describes that the bisporphyrin cleft molecule with two borate centers adopts the (P)- and (M)-helical conformations where the chiral guests are encapsulated with fairly good diastereoselectivities. The interconversion between the (P)- and (M)-helical conformations requires very high activation energy allowing the isolation of them at room temperature, but the interconversion is turned on upon the addition of water that most likely breaks the tetrahedral spiroborate structures in the host molecule. The diastereoselectivities of the bisporphyrin cleft for the chiral guests are fairly high but are influenced by the solvent systems. Based on the careful consideration of the complex structures using NMR techniques, the authors conclude that the CH- π interaction between the host-guest complex determines the stereo selection; thereby, the competitive solvent against the CH- π interaction reduces the selectivities. The manuscript is well-written, and the detailed discussion and experiments result in the solid conclusion. In my opinion, this manuscript should be publishable in *Nature Communication* after justifying the following comments.

-In pg 4, the authors describe "the inclusion complex formation of (M)-1 with G1 resulted in an

increase in the racemization rates of (M)-1•G1 as compared to those of the free (M)-1, probably due to the B-O bond strain at the spiroborate moieties, being increased by the G1 complexation." But, I think of the guest complexation resulted in the huge stabilization with a DG of -107 kJ/mol in acetonitrile, which should increase the activation energy if assuming the TS energy is not affected upon the complexation. If the author's hypothesis might be correct, the strain of the spiroborate moieties should significantly increase the free energy change of the complexation, breaking the complex. I think of the TS of the complex don't resemble the TS of 1. The authors should describe the reason why the reaction rate is accelerated upon the guest complex in detail. -In ESI, the binding constants were determined by FL quenching experiments. But, dynamic and static mechanisms are involved in the quenching process. Stern-Volmer plots should be evaluated for the complexation. In general, the significant contribution of the dynamic quenching process results in an overestimation of a binding constant. When the static quenching process (molecular association) is dominant, Stern-Volmer plot curves upward. The equation given in ESI appears not to evaluate the dynamic quenching process. The authors should reevaluate the binding constants that were determined using the equation given.

Reviewer #3 (Remarks to the Author):

The manuscript is of excellent level and there is nothing to say about the quality of data and their interpretation. However, I do not think that the results have all the various aspects (novelty, urgency etc.) required by a publication on Nature Communications. I cannot recommend publication of this manuscript.

Responses to Reviewer #1:

Comment 1: I find it hard to evaluate the novelty of this work. The manuscript claims the process of deracemization and specifically the use of water as the activator of this process to be the original element of this work. They make further reference to the double helix as a shape related to DNA and therefore glean the luster coming from all of DNA's biological implications.

What I like is the complexity of the system and that it seems to work efficiently. There is something appealing about these kinds of structures.

Reply 1: We appreciate this positive and encouraging comment.

Comment 2: That said, what I don't like is the justification. Deracemization as a process is as old as basic 1st and 2nd order asymmetric transformations. That these occur selectively in the presence of chiral agents was postulated at least as early as Quehl and Pfeiffer's work on chiral coordination complexes in *Berichte*, 1931, 64, 2667 and by Kuhn for biaryls in *Berichte* 1932 65, 49. These cases were of labile systems shifted to one form, but the degree of lability could be easily imagined as tunable and perhaps has been in the intervening 85 years.

Norden showed in the 1986 the deracemization of iron complexes on DNA, see: Hård T, Nordén B. Enantioselective interactions of inversion-labile trigonal iron(II) complexes upon binding to DNA. *Biopolymers*. 1986 Jul;25(7):1209-28. Thus the idea of using DNA or double-helix forms has been broached.

So I don't see the authors as having made a milestone leap in the concept of deracemization, but perhaps the peculiarity of borate esters as the stability element and water and the tunable factor make this of special interest to readers.

At a minimum, I would hope the authors would teach the reader more about 1st order asymmetric transformations and their history dating back a century. I would ask that they down play statements claiming that deracemizations is somehow new or only applied to very few systems. Indeed, they should highlight that the energy difference only has to be a couple of kcal/mol so in polymers this might be even easier to achieve. One a proper context is laid out, then their novel water switching mechanism will probably tickle some reader's fancy.

Reply 2: We appreciate these important comments and suggestions. Accordingly, we properly revised our manuscript by adding brief introduction of deracemization (pages 2-3) with the following references listed above by this reviewer. The changes are highlighted in a copy of our revised manuscript including revised Supplementary Information as Review-Only Supplementary Information (Yashima-ROSI.pdf) marked with a red color.

As ref. 24: Pfeiffer, P. & Quehl, K. Über einen neuen effekt in Lösungen optisch-aktiver substanzen (I. Mitteil). *Ber. Dtsch. Chem. Ges.* **64**, 2667–2671 (1931).

As ref. 25: Werner, A. Über spiegelbild isomerie bei chromverbindungen. III, *Ber. Dtsch. Chem. Ges.* **45**, 3061–3070 (1912).

As ref. 27: Kuhn, R. Über einen neuen effekt in lösungen optisch-aktiver substanzen. *Ber. Dtsch. Chem. Ges.* **65**, 49–51 (1932).

As ref. 28: Härd, T. & Nordén, B. Non-sequential processes for the transformation of a racemate into a single stereoisomeric product: proposal for stereochemical classification. *Biopolymers*, **25**, 1209-1228 (1986).

In relation to this revision, we newly added the following references on deracemization to the revised manuscript in order for the readers to understand the novelty of our work more clearly.

As ref. 22: Eliel, E. L., Wilen, S. H. & Mander, L. N. *Stereochemistry of Organic Compounds* 1st ed., chap. 7 (John Wiley & Sons, 1994).

As ref. 26: Bosnich, B. Asymmetric syntheses, asymmetric transformations, and asymmetric inductions in an optically active solvents. *J. Am. Chem. Soc.* **89**, 6143-6148 (1967).

As ref. 29: Lacour, J. & Frantz, R. New chiral anion mediated asymmetric chemistry. *Org. Biomol. Chem.* **3**, 15-19 (2005).

As ref. 30: Green, M. M., Khatri, C. & Peterson, N. C. A macromolecular conformational change driven by a minute chiral solvation energy. *J. Am. Chem. Soc.* **115**, 4941-4942 (1993).

As ref. 33: Faber K. Non-sequential processes for the transformation of a racemate into a single stereoisomeric product: Proposal for stereochemical classification. *Chem. Eur. J.* **7**, 5004-5010 (2001).

As ref. 37: Rachwalski, M., Vermue, N. & Rutjes, F. P. J. T. Recent advances in enzymatic and chemical deracemization of racemic compounds. *Chem. Soc. Rev.* **42**, 9268-9282 (2013).

As ref. 38: Amabilino, D. & Kellogg, R. M. Spontaneous Deracemization. *Isr. J. Chem.* **51**, 1034-1040 (2011).

We also newly added the following reference on spiroborate-based helicates that is not related to the present bisporphyrin-based helicate.

As ref. 41: Ousaka, N. *et al.* Spiroborate-based double-stranded helicates: *Meso-to-racemo* isomerization and ion-triggered springlike motion of the *racemo*-helicate. *J. Am. Chem. Soc.* **140**, 17027-17039 (2018).

Responses to Reviewer #2:

Comment 1:

The manuscript by Yashima et al. describes that the bisporphyrin cleft molecule with two borate centers adopts the (P)- and (M)-helical conformations where the chiral guests are encapsulated with fairly good diastereoselectivities. The interconversion between the (P)- and (M)-helical conformations requires very high activation energy allowing the isolation of them at room temperature, but the interconversion is turned on upon the addition of water that most likely breaks the tetrahedral spiroborate structures in the host molecule. The diastereoselectivities of the bisporphyrin cleft for the chiral guests are fairly high but are influenced by the solvent systems. Based on the careful consideration of the complex structures using NMR techniques, the authors conclude that the CH- π interaction between the host-guest complex determines the stereo selection; thereby, the competitive solvent against the CH- π interaction reduces the selectivities. The manuscript is well-written, and the detailed discussion and experiments result in the solid conclusion. In my opinion, this manuscript should be publishable in Nature Communication after justifying the following comments.

Reply 1: We appreciate this positive and encouraging comment.

Comment 2: In pg 4, the authors describe “the inclusion complex formation of (M)-1 with G1 resulted in an increase in the racemization rates of (M)-1•G1 as compared to those of the free (M)-1, probably due to the B-O bond strain at the spiroborate moieties, being increased by the G1 complexation.” But, I think of the guest complexation resulted in the huge stabilization with a DG of -107 kJ/mol in acetonitrile, which should increase the activation energy if assuming the TS energy is not affected upon the complexation. If the author’s hypothesis might be correct, the strain of the spiroborate moieties should significantly increase the free energy change of the complexation, breaking the complex. I think of the TS of the complex don’t resemble the TS of 1. The authors should describe the reason why the reaction rate is accelerated upon the guest complex in detail.

Reply 2: We appreciate this invaluable comment. We performed the racemization rate measurements of the bisporphyrin helicate (M)-1 in the presence and absence of G1 guest three times (Supplementary Table 1), and we confirmed its reproducibility. We agree with the reviewer’s comment that the complex formation of (M)-1 with an electron-deficient aromatic guest G1 stabilized its inclusion complex through favorable face-to-face stacking interactions between the bisporphyrin and G1 under thermodynamic control, which, however, resulted in remarkable structural changes; the distance between the porphyrin rings expanded from 4.1 to 6.8 Å, causing steric strain in the spiroborate helix, which could be alleviated by a unidirectional rotation of the porphyrin rings and a twisting of the helicate in one direction (see ref. 48). The free helicate (M)-1 and its inclusion complex (M)-1_{TBAi}⊃G1 are kinetically inert toward racemization in the absence of water. However, in the presence of water, such steric strain within the spiroborate helicate could be relaxed during the water catalyzed B–O bond cleavage reactions at the spiroborate groups. As a result, the (M)-1_{TBA2}⊃G1 inclusion complex racemized faster than the free (M)-1 did, leading to a significant increase in the activation entropy (ΔS^\ddagger) for the racemization from -42 ± 11 ((M)-1) to 4 ± 20 J mol⁻¹ ((M)-1⊃G1) (Supplementary Table 1) as described in the manuscript. Intermolecular hydrogen bond formation between the carboxylic anhydride residues of

G1 included in the helicate and water may also contribute to the observed acceleration of the water-catalyzed racemization of the (*M*)-**1**_{TBA2}⊃**G1** inclusion complex. Thus, the following sentences “Such steric strain within the spiroborate helicate could be relaxed during the water catalyzed B–O bond cleavage reactions at the spiroborate groups, leading to a significant increase in the activation entropy (ΔS^\ddagger) for the racemization from -42 ± 11 ((*M*)-**1**_{TBA2}) to 4 ± 20 J mol⁻¹ ((*M*)-**1**_{TBA2}⊃**G1**) (Supplementary Table 1). Intermolecular hydrogen bond formation between the carboxylic anhydride residues of **G1** and water, which may accelerate the water-catalyzed racemization of the (*M*)-**1**_{TBA2}⊃**G1** inclusion complex, could also be taken into consideration.” have been added in the revised manuscript (page 4, lines 2-8 from the bottom).

Comment 3: In ESI, the binding constants were determined by FL quenching experiments. But, dynamic and static mechanisms are involved in the quenching process. Stern-Volmer plots should be evaluated for the complexation. In general, the significant contribution of the dynamic quenching process results in an overestimation of a binding constant. When the static quenching process (molecular association) is dominant, Stern-Volmer plot curves upward. The equation given in ESI appears not to evaluate the dynamic quenching process. The authors should reevaluate the binding constants that were determined using the equation given.

Reply 3: According to the suggestion, the Stern-Volmer plots of (*M*)-**1**_{TBA2} quenched by (*S*)- and (*R*)-**G2** in diluted THF (1.0 μM) and CH₃CN (0.50 μM) solutions measured at 25 °C are shown in Supplementary Figs. 10e and 11e, respectively, in the revised Supplementary Information. The linear plots obtained by the least-squares curve fitting method indicated the static quenching processes of (*M*)-**1**_{TBA2} by (*S*)- and (*R*)-**G2** in THF and CH₃CN that were dominant as anticipated from the ¹H NMR spectra, in which the signals due to free and complexed **G2** were separately observed because of slow exchange between them (Supplementary Fig. 12d and e, respectively).

Responses to Reviewer #3:

Comment 1: The manuscript is of excellent level and there is nothing to say about the quality of data and their interpretation.

Reply 1: We appreciate this positive and encouraging comment.

Comment 2: However, I do not think that the results have all the various aspects (novelty, urgency etc.) required by a publication on Nature Communications. I cannot recommend publication of this manuscript.

Reply 2: As described in the manuscript (Abstract and Introduction sections), deracemization is a powerful method by which a racemic mixture can be transformed into an excess of one enantiomer with the aid of chiral auxiliaries, but has been applied only to small chiral molecular systems. What we would like this reviewer to understand the novelty of our work is that the present study reports the first deracemization of a racemic helicate, which is kinetically stable (*static*). In contrast to the reported examples of deracemization, the helicate possesses both dynamic (labile) and static (inert) features toward racemization in the presence and absence of water, respectively, and eventually undergoes deracemization upon inclusion complexation with a chiral guest triggered by water, thus producing an optically-active *static* helicate with a high enantioselectivity, after the removal of water. Furthermore, we successfully revealed stereospecific CH- π interactions which play a key role in the diastereoselective inclusion complex formations between the helicate and chiral guests, leading to a highly helix-sense selective deracemization of the helicate, as supported by detailed NMR measurement results along with quantum chemical calculations. Therefore, we believe that the scientific quality, novelty and impact of our work are incomparably high. Nevertheless, we have revised our manuscript (pages 2-3, Introduction section) as highlighted in a copy of our revised manuscript including revised Supplementary Information as Review-Only Supplementary Information (Yashima-ROSI.pdf) marked with a red color in order for the readers to follow our results much easier than before and to understand the novelty of our work even more clearly.

REVIEWERS' COMMENTS:

Reviewer #1 (Remarks to the Author):

The authors have done a nice job expanding the context of manuscript. They have also been able to set their work out from the pack and made the case for publishing in Nature Communications stronger. I support publication.

Reviewer #2 (Remarks to the Author):

I am satisfied that the authors have properly responded to the critiques in the revision. The manuscript is now suitable for publication.

Reviewer #3 (Remarks to the Author):

Dear Editor,

I am in a quite tricky situation. My doubts on the manuscript or not on the scientific side; I find both data and interpretation robust. My doubt concerns the matching between the journal and the manuscript. It is quite difficult to change mind on this. On the other hand I would like not to write again the same criticisms. So please consider my position as neutral and if the other two reviewers are positive please proceed with publication. For this reason I will not send you any comment.